# MULTI-TASK PRETRAINING DRIVES REPRESENTATIONAL CONVERGENCE

## ABSTRACT

What determines the geometry of a neural network's internal representations, and when do different training objectives lead to the same representational solution? We study these questions using a controlled framework in which small transformers are trained on geometric tasks defined over real-world city coordinates. We find that single-task training produces diverse, task-specific representational geometries, from thread-like structures to 2D manifolds to fragmented clusters. However, multi-task training drives rapid representational convergence: models trained on different task combinations develop increasingly similar internal representations, as measured by CKA. A 7-task model spontaneously recovers world-map-like structure in raw PCA; while linear world representations exist in all models, multi-task training amplifies their magnitude until they dominate the principal components. These findings provide controlled evidence for the Multitask Scaling Hypothesis, one proposed mechanism underlying the Platonic Representation Hypothesis.

## 1 INTRODUCTION

Neural networks can develop structured internal representations that capture meaningful aspects of their training data (Bengio et al., 2014), including geographic, temporal, and relational structure in language models (Li et al., 2022; Gurnee & Tegmark, 2023; Nanda et al., 2023b). The Platonic Representation Hypothesis (PRH) posits that models trained on diverse data converge toward similar representations (Huh et al., 2024).

However, key questions remain: what determines representation geometry, and when do different objectives yield the same solution? These are difficult to study in real-world settings where world, data, and model are entangled.

**This work.** To study these questions, we decouple the underlying *world* from the *data generation process* to control them independently. Concretely, we adopt the coordinates of real-world cities as our "world," a ready-made complex structure with ground-truth geometry, and define 7 geometric tasks on top of it. We train autoregressive Transformers on this data and study how world representations form and vary across tasks, surfacing preliminary evidence for the Platonic Representation Hypothesis (PRH) (Huh et al., 2024). Our main contributions are:

- **A Framework Decoupling World, Data and Model.** We separate the underlying world (city coordinates) from the data generation process (7 geometric tasks), enabling systematic study of how different tasks shape representations of the same world. The world provides ground-truth coordinates for directly assessing representation quality via probing.

- **Task-Dependent Geometry and Multi-Task Convergence.** We show that different tasks operating on the same world produce highly variable representational geometries across tasks and seeds. However, multi-task training drives convergence: models trained on multiple tasks show higher representational alignment, even when they share no common tasks. This provides partial evidence for the Multitask Scaling Hypothesis, one proposed mechanism for the Platonic Representation Hypothesis.

For related work, see App. A.

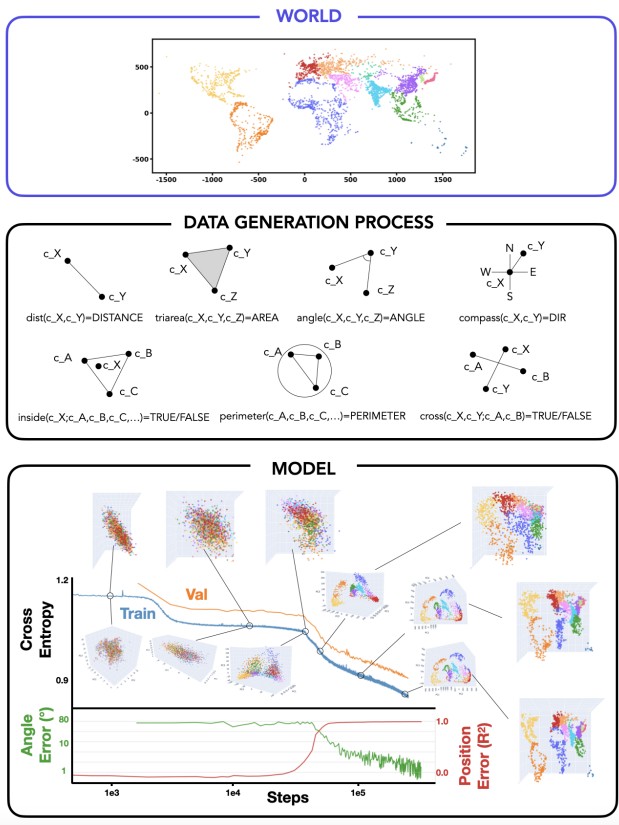

Figure 1: **Overview of the World-Data-Model framework. Top:** The world consists of 5,075 real city coordinates (App. C.1). **Middle:** Seven geometric tasks generate training data from city coordinates (App. C.2). **Bottom:** Training dynamics of one model, showing loss curves, linear probing accuracy for coordinate reconstruction and visualizations of internal representations (PCA and linear probe projections) at different training stages. See App. Fig. 6 for all training curves.

## 2 RESULTS

We study how task composition shapes internal world representations by training small transformers on geometric tasks over 5,075 real-world city coordinates (Fig. 1). Models never see coordinates directly, only task queries like `dist(c_0865,c_4879)=769`, allowing us to probe whether they internally reconstruct world structure. See App. C for full experimental details.

**Result 1: World Representations Form Early and Freeze** World representations appear to crystallize early in training and then stop changing (Fig. 1, bottom). Training on the `angle` task, the model starts with random representations, goes through a loss plateau while clustering nearby cities, then forms world-aligned geometry as loss drops. Once this structure forms (roughly within the first ~15% of training), it remains essentially frozen, even as loss continues to decrease and task accuracy improves over the remaining 85% of training (see App. Fig. 7 for visualizations across tasks). The linear probe $R^2$ for coordinate decoding rises slightly before task accuracy, reminiscent of hidden progress measures in grokking (Nanda et al., 2023a). Overall, we observe that autoregressive training reliably produces world representations, but the window for shaping them appears brief. This sets the stage for our main question: how do different tasks shape these representations?

**Result 2: Data Generation Process Controls World Representation Geometry** We train models from scratch for each of the seven tasks and visualize their representations in Fig. 2(a): PCA projections, linear probe reconstructions and rotated views.

Different tasks produce qualitatively distinct geometries: `distance` forms thread-like structures, `angle` forms 2D manifolds, `compass` forms fragmented clusters, and `inside` forms diffuse rep-

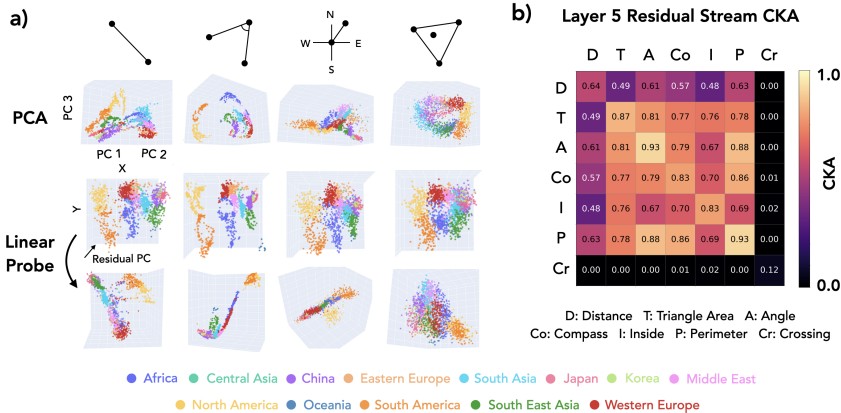

Figure 2: **World representation geometry depends on the data generation process.** (a) Different tasks create distinct geometries: `distance` (thread-like), `angle` (2D manifold), `compass` (fragmented), `inside` (diffuse). Row 1: PCA. Row 2: Linear probe projections. Row 3: Rotated views showing hidden structure. See App. Fig. 8 for more seeds. (b) CKA matrix at layer 5, estimated across 3 seeds. `Crossing` (Cr) fails to train alone. See App. Fig. 9 for SEM and layers 3, 4, 6. 3D visualizations: link .

resentations. These qualitative patterns are relatively consistent across random seeds (see App. E.2). Despite geometric differences, we can linearly decode (x,y) coordinates from most tasks (row 2), though some tasks (`angle`) yield cleaner reconstructions than others. The third row shows manually rotated views revealing that representations differ substantially in non-probe directions, a reminder that *linear probing only surfaces what we look for.*

We quantify representational similarity using CKA (Kornblith et al., 2019) (Fig. 2b). We find substantial variability even across seeds for the same task (see App. Fig. 9), but cross-task differences remain clear: `distance` produces particularly divergent representations, a result not obvious from PCA visualizations or from intuition about the task. Note: the `crossing` task fails to train in isolation[1], explaining its near-zero CKA; intriguingly, it succeeds in multi-task settings (Result 3).

**Result 3: Multi-Task Learning Drives Representational Convergence**  Having established that single-task training produces variable representations, we now ask: does multi-task training reduce this variability? This question partially connects to PRH (Huh et al., 2024), which observes that neural networks trained on diverse data develop aligned representations even across different modalities and architectures. One potential mechanism they suggest is the Multitask Scaling Hypothesis:

> *"There are fewer representations that are competent for N tasks than there are for M ≤ N tasks. As we train more general models that solve more tasks at once, we should expect fewer possible solutions."*

Our setup provides a potential testbed for this hypothesis, with a ground-truth world model and multiple tasks defined over it. We trained models on selected two-task combinations (3 seeds each; see App. Fig. 12 for all 21 combinations). Fig. 3(a) shows representations when trained jointly on `distance` and `triangle area` (with single-task models shown for comparison), while (b) shows `inside` and `perimeter`. When trained on two tasks, models develop more regular representational structures. While difficult to appreciate in static 2D projections, we encourage readers to explore our interactive 3D visualizations at this link .

We measure CKA between two-task trained models to quantify this alignment (Fig. 3(c)). CKA is substantially higher than for single-task models. One might expect high CKA when models share a task, but even models trained on completely disjoint task pairs show substantially higher alignment. In Fig. 3(d), we plot average CKA for models trained on 1, 2, and 3 tasks across layers 3-6,

---

[1] This likely connects to known hard-to-learn dynamics and gradient plateaus in training transformers (Pezeshki et al., 2021; Shah et al., 2020; Hoffmann et al., 2024; Bachmann & Nagarajan, 2025; Gopalani & Hu, 2025).

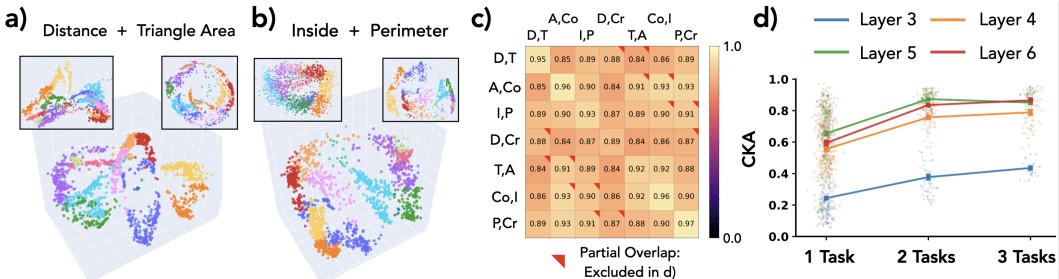

Figure 3: **Multi-task pretraining drives representational convergence.** (a,b) Two-task training creates more regular structures than single-task models. (c) CKA matrix (7×7) for two-task models shows higher alignment (see App. Fig. 10 for SEM). (d) Average CKA increases with task count (1→2→3), saturating at ∼0.85 for layers 4-6 while layer 3 continues improving (see App. Fig. 11 for SEM). `Crossing`, which failed to learn in single-task training, is excluded; including it would only strengthen the convergence finding.

averaging only over models with completely disjoint task sets. Training on more tasks clearly leads to more aligned representations, with CKA saturating around 0.85 for 2 and 3 tasks in layers 4-6, while layer 3 continues improving. Notably, multi-task training also reduces per-seed variance of representations (App. Fig. 12b).

Overall, we find that *multi-task learning leads to more aligned model internal representations*, providing partial evidence for the Multitask Scaling Hypothesis explanation of PRH.[2] Crucially, this alignment emerges even though single-task models achieve comparable task performance (all models reach high accuracy on their respective tasks). Since our networks are trained to representational convergence (as seen in Fig. 1), this demonstrates that the alignment is not simply a byproduct of optimization difficulty but rather that task diversity, not just data quantity or performance pressure, drives aligned representation learning.

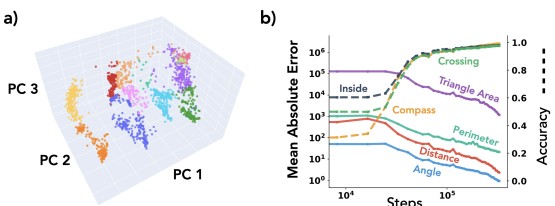

Figure 4: **7-task model.** (a) PCA projection of layer 5 representations naturally reveals world map structure. (b) Training curves showing successful learning of all 7 tasks, including `crossing` which failed in single-task training.

An auxiliary finding: the `crossing` task, which was unlearnable alone, trains successfully when paired with any other task. We speculate that companion tasks provide structured coordinate representations that `crossing` can leverage, an implicit curriculum where easier tasks scaffold the learning of harder ones through shared representations.

To extend these findings, we trained a model on all 7 tasks simultaneously (Fig. 4). This model successfully learns all tasks. Strikingly, *raw PCA of this model's representations directly reveals world map structure*, without any linear probe. Recall that for single-task models, we needed to train a linear probe to decode coordinates (Fig. 2, row 2); the world was hidden in the representations. For the 7-task model, the world simply appears in the first two principal components. While linear world representations exist in all trained models (we can always decode coordinates via probing), multi-task training appears to amplify their magnitude until they dominate the variance structure.

## 3 CONCLUSION

Our results provide controlled evidence for the Multitask Scaling Hypothesis of PRH: task diversity drives representational convergence, even across models trained on completely disjoint task sets.

---

[2]A full test of PRH would require showing convergence across different architectures; we test only the task-diversity mechanism here.

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

# APPENDIX

## A  RELATED WORK

**Internal Representations.**  Understanding internal representations has roots in neuroscience (Hubel & Wiesel, 1962), informing early neural network development (Fukushima, 1980; Bengio et al., 2014; Rosenblatt, 1958; Rumelhart et al., 1986). Recent work has revealed that language models develop structured "world models" encoding geographic, temporal and relational information (Li et al., 2022; Gurnee & Tegmark, 2023; Nanda et al., 2023b; Marks & Tegmark, 2024), with similar representations emerging during in-context learning (Vafa et al., 2025). Mechanistic interpretability and sparse autoencoders have enabled decomposition of neural activations into interpretable features (Anthropic AI, 2023; Templeton et al., 2024). Researchers have also uncovered that models represent meaningful properties of data, including concepts (Pearce et al., 2025; Higgins et al., 2017), features (Olah et al., 2017), and abstractions (Lee et al., 2025; Arditi et al., 2024), in interpretable ways. Furthermore, PRH posits that diverse models converge toward similar representational structures (Huh et al., 2024). However, recent work questions this representational optimism, suggesting that deep network representations may be more brittle than previously assumed (Kumar et al., 2025). Only recent work has begun examining how representations emerge during pretraining in real LLMs (Li et al., 2025; Ge et al., 2025). Our work takes a complementary perspective, studying the factors that control the formation of these representations during pretraining.

**Dynamics of Representations.**  Recent work has begun studying how representations evolve during in-context learning (Shai et al., 2025; Demircan et al., 2024) or fine-tuning (Casademunt et al., 2025; Minder et al., 2025). Relatedly, Lubana et al. (2025) show that representations exhibit rich temporal dynamics that standard interpretability methods (e.g., SAEs) fail to capture due to stationarity assumptions. Fu et al. (2025) show that VLMs trained by merging LLMs and vision encoders often fail to utilize representations surfaced by the vision encoder, i.e. the representations exist but remain unused.

**Geometric Deep Learning.**  Geometric deep learning studies how data geometry interacts with model architectures, developing equivariant networks that respect symmetries (Bronstein et al., 2021; Cohen & Welling, 2016; Weiler & Cesa, 2021). While our world is defined on a 2D plane, one might ask: why not a sphere, torus, or other manifold? This is an interesting direction, but not our focus. We study how neural networks adapt internal representations to tasks in an arbitrarily chosen geometry. Moreover, a change in world geometry can be absorbed into the task definition (e.g., geodesic vs. Euclidean distance), so the key question remains how representations form given the task, not the underlying manifold. Planar coordinates also allow clean linear probing of world representations. Our models are standard transformers without geometric priors; we study what representations emerge purely from training on task data, treating geometry as emergent rather than imposed.

**Loss Plateaus.**  Our `crossing` task fails to learn in single-task training despite escaping an initial plateau (likely output format learning), suggesting it remains stuck in a deeper plateau. Such plateaus are notoriously difficult for transformers. Recent work has studied this phenomenon mechanistically in transformers (Hoffmann et al., 2024; Gopalani & Hu, 2025; Singh et al., 2024), while others relate it to more general optimization challenges in deep learning such as simplicity bias and gradient starvation (Shah et al., 2020; Pezeshki et al., 2021; Bachmann & Nagarajan, 2025). Most related to our findings, Kim et al. (2025) show that multi-task training shortens loss plateaus, similar to why our `crossing` task trains successfully when joined with any other task.

## B  3D VISUALIZATIONS

3D visualizations are available here (Open Science Framework link).

## C   Experimental Details

This section provides detailed information about the world, data generation process, model architecture, and training procedures used in our experiments.

### C.1   World

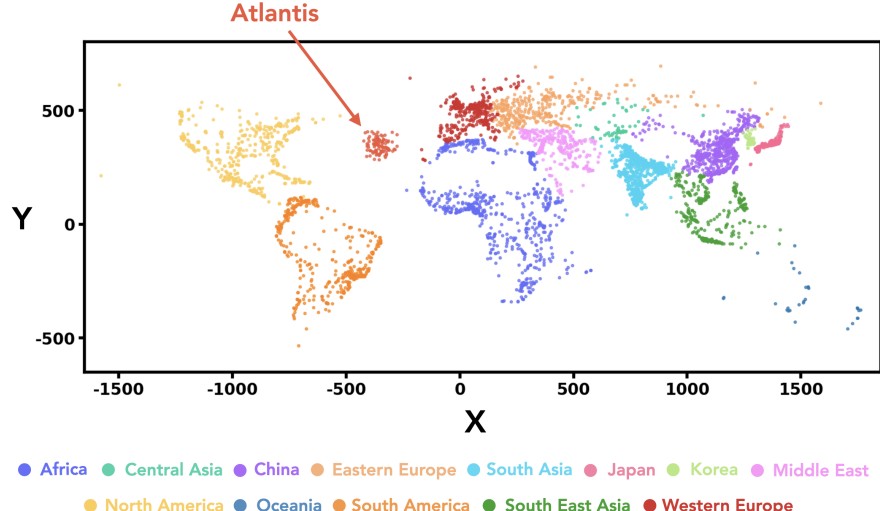

Figure 5: **Geographic distribution of cities used in our experiments.** 5,075 real-world cities spanning all continents, providing a fixed, measurable world structure. Coordinates use an equirectangular projection: $x = 10 \times$ longitude, $y = 10 \times$ latitude (in degrees).

Our experiments use a geographic world consisting of 5,075 cities extracted from the GeoNames (OpenDataSoft / GeoNames, 2025) database with population greater than 100,000. Cities are distributed across all continents. This choice provides natural variation in density (e.g., dense regions like India versus sparse Oceania) that creates interesting computational challenges.

While we use real city coordinates, this work studies abstract geometric reasoning rather than actual geography. We project coordinates to Euclidean space using an equirectangular projection (as described above) and treat all tasks as pure geometry problems.

We deliberately chose a flat 2D manifold rather than a spherical globe. Our early experiments used spherical coordinates, but we realized that regardless of the external world's geometry, the model must construct its own internal representation starting from random entity distributions. Given the model's nonlinearity, there is no fundamental reason why any particular geometry (planar, spherical, etc.) would be canonical. Our choice of planar geometry enables clean linear probing to read out world representations, whereas extracting nonlinear manifold structure remains an open challenge (Engels et al., 2024; Csordás et al., 2024). While geometric deep learning (Bronstein et al., 2021) studies the interaction between data geometry and model computation, our focus is on general sequence modeling rather than geometry-aware architectures.

City IDs are randomly assigned from the range [0, 9999], creating a sparse identifier space that models must learn to map to coordinates. All coordinates are stored as integers (after the $\times 10$ scaling), eliminating floating-point precision issues.

### C.2   Data Generation Process

**Tasks**   We implement 7 geometric tasks that operate on city coordinates. All tasks use a consistent format: `task(arguments)=answer`, where city IDs are prefixed with `c_`. Numerical outputs (distance, area, angle, perimeter) are rounded to integers. Table 1 summarizes the tasks.

| Task | Input | Output Type | Unit/Values | Example |
|------|-------|-------------|-------------|---------|
| distance | 2 cities | Numerical | Scaled coords | `dist(c_865,c_4879)=769` |
| triarea | 3 cities | Numerical | Scaled coords$^2$ | `triarea(c_1234,c_5678,c_9012)=45823` |
| angle | 3 cities | Numerical | Degrees (0–180) | `angle(c_2345,c_6789,c_123)=97` |
| compass | 2 cities | Categorical | 8 directions | `compass(c_1234,c_5678)=NE` |
| inside | $1 + n$ cities | Categorical | TRUE/FALSE | `inside(c_9012;c_3456,...)=FALSE` |
| perimeter | $n$ cities | Numerical | Scaled coords | `perimeter(c_4567,c_8901,...)=2856` |
| crossing | 4 cities | Categorical | TRUE/FALSE | `cross(c_2345,c_6789;c_123,c_4567)=TRUE` |

Table 1: Summary of 7 geometric tasks. Numerical outputs are integers; "scaled coords" refers to the $\times 10$ coordinate system (Sec. C.1). Categorical tasks have discrete outputs: compass uses 8 cardinal directions (N, NE, E, SE, S, SW, W, NW), while inside and crossing are binary. The inside task tests if the first city lies within the convex hull of the remaining cities; crossing tests if line segment $(c_1, c_2)$ intersects segment $(c_3, c_4)$.

**Dataset Sizes**   Each pretraining set consists of 1M rows of data per task. For multi-task pretraining, we combine datasets from multiple tasks (e.g., 7M rows for the 7-task model).

### C.3 MODEL AND TRAINING

**Tokenization**   We use character-level tokenization with 98 ASCII tokens (excluding space, which serves as the delimiter), plus special tokens for beginning-of-sequence (BOS), end-of-sequence (EOS), and padding (PAD). Each task query and answer is tokenized character-by-character (e.g., `dist(c_0865,c_4879)=769` becomes `d i s t ( c _ 0 8 6 5 , c _ 4 8 7 9 ) = 7 6 9`).

This character-level scheme is intentional. While assigning each city and task a dedicated token would simplify learning, such synthetic-friendly tokenization does not reflect how real language models operate. LLMs must handle multi-token entities, variable-length prompts (our task prefixes have different lengths), computations at different sequence positions, and irregularly tokenized content (e.g., numbers in LaTeX). Preliminary experiments exploring pitfalls of next-token prediction (Bachmann & Nagarajan, 2025) showed that tokenization details qualitatively affect results. We therefore chose character-level tokenization to better approximate realistic sequence modeling conditions.

**City ID Assignment**   City IDs are randomly assigned from the range $[0, 9999]$, ensuring no geographic information leaks through the identifier. This random assignment means the model cannot exploit ID patterns to infer coordinates.

**Architecture**   We use the Qwen2 (Yang et al., 2024) decoder-only transformer architecture with hidden size 128, 4 attention heads, and 6 layers.

**Pretraining**   We train models autoregressively on the full sequence (no prompt masking). While we observed training speedup when masking loss computation on the prompt side, we deliberately avoid this optimization to maintain similarity with standard autoregressive language model pretraining. All pretraining runs see 42M rows regardless of dataset size (e.g., 42 epochs for 1M rows, 6 epochs for 7M rows). Table 2 summarizes the hyperparameters.

## D   ANALYSIS METHODS

### D.1 EVALUATION

**Generation Protocol**   For evaluation, we use teacher forcing up to the "=" sign (the prompt), then generate autoregressively at temperature zero until reaching the EOS token or a maximum of 128 tokens (sufficient for all tasks). All trained models achieve perfect parse accuracy: outputs always match the expected format (integers for numerical tasks, valid categories for categorical tasks).

| Hyperparameter | Value |
|---|---|
| Optimizer | AdamW (Loshchilov & Hutter, 2019) |
| Learning rate | $3 \times 10^{-4}$ |
| Weight decay | 0.01 |
| Scheduler | Linear with warmup |
| Warmup steps | 50 |
| Batch size | 128 |
| Max sequence length | 256 |
| Total training rows | 42M |
| Initialization scale | 0.1 (std) |

Table 2: **Pretraining hyperparameters.**

**Task-Specific Metrics** Categorical tasks (`compass`, `inside`, `crossing`) are evaluated using accuracy. Numerical tasks are evaluated using absolute error: `distance` (scaled coordinate units), `triarea` (scaled coordinate units$^2$), `angle` (degrees), and `perimeter` (scaled coordinate units).

## D.2 REPRESENTATION EXTRACTION

We extract representations from the residual stream after transformer blocks, specifically at layers 3, 4, 5, and 6 of our 6-layer model. Unless otherwise specified, all representation analyses in this paper use layer 5 representations.

To extract city representations, we pass a task prefix followed by a city ID through the model. For single-task models, we use the corresponding task prefix. For multi-task models (2-task and 3-task), we use the first task in the combination as the prefix. We verified that the choice of task prefix has negligible effect on the extracted city representations.

For a city with ID 1234, the input sequence is:

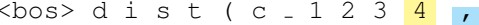

We extract and concatenate the representations of two tokens: (1) the last digit of the city ID and (2) the following delimiter token (typically a comma). This yields a 256-dimensional representation ($128 \times 2$) per city, which we use for both PCA visualization and linear probing.

**Omitting cities with leading zeros** We omit cities with IDs starting with 0, 00, or 000 from representation analyses. These cities form distinct clusters in representation space, separate from cities with IDs starting with non-zero digits. We hypothesize this occurs because the digit 0 has special semantic status: in numerical outputs (distances, angles, areas), leading zeros never appear (e.g., "=769" not "=0769"), so the model learns to treat 0 differently when it appears as a leading digit. When 0 appears at the start of a city ID, the model may encode a feature indicating "this is an identifier, not a number," causing these cities to cluster separately. To ensure consistent evaluation across all cities, we exclude IDs matching the pattern `^[0][0-9]*$` (i.e., any ID starting with zero).

## D.3 LINEAR PROBING & PCA

We use the representations described in Sec. D.2 for both PCA visualization and linear probing.

**Linear Probing** We train linear probes to predict city coordinates $(x, y)$ from the 256-dimensional representations. We use a train/test split of 3250/1250 cities, training separate probes for $x$ and $y$ coordinates via ordinary least squares (OLS) without regularization. We report $R^2$ scores and mean absolute error in scaled coordinate units.

**PCA** For visualization, we apply PCA to the representations and plot the first two or three principal components. We use consistent color coding based on geographic region to enable visual comparison across models and seeds.

### D.4 CENTERED KERNEL ALIGNMENT

We use Centered Kernel Alignment (CKA) (Kornblith et al., 2019) to measure representational similarity between models. Given two representation matrices $X \in \mathbb{R}^{n \times d_1}$ and $Y \in \mathbb{R}^{n \times d_2}$ (same $n$ cities, potentially different dimensions), we compute linear kernel matrices $K = XX^T$ and $L = YY^T$, center them, and compute:

$$\text{CKA}(X, Y) = \frac{\langle K, L \rangle_F}{\|K\|_F \|L\|_F} \tag{1}$$

where $\langle \cdot, \cdot \rangle_F$ denotes the Frobenius inner product. CKA yields a similarity score in $[0, 1]$ that is invariant to orthogonal transformations and isotropic scaling.

For each pair of models, we extract city representations (Sec. D.2) and compute CKA between the resulting matrices. We filter cities to exclude IDs starting with zeros (see Sec. D.2). We report CKA values at layers 3, 4, 5, and 6, with layer 5 as the default unless otherwise specified.

## E ADDITIONAL EXPERIMENTS & RESULTS

### E.1 TRAINING DYNAMICS

Fig. 6 shows training dynamics for all seven single-task models. Each panel displays three rows of metrics over gradient steps: (top) training and validation loss, (middle) task-specific performance metric alongside linear probe $R^2$ for coordinate decoding, and (bottom) linear probing distance error measuring how accurately city coordinates can be reconstructed from representations.

Several patterns emerge across tasks. First, all tasks except `crossing` eventually achieve high coordinate $R^2$ (red curves reaching $\sim 1.0$), indicating that world representations form reliably across diverse geometric objectives. Second, the relationship between loss, task performance, and coordinate decodability varies across tasks. Third, `crossing` (panel g) fails entirely in single-task training. Loss remains high, accuracy stays near chance, and coordinate $R^2$ never rises, consistent with the main text observation that this task requires multi-task scaffolding.

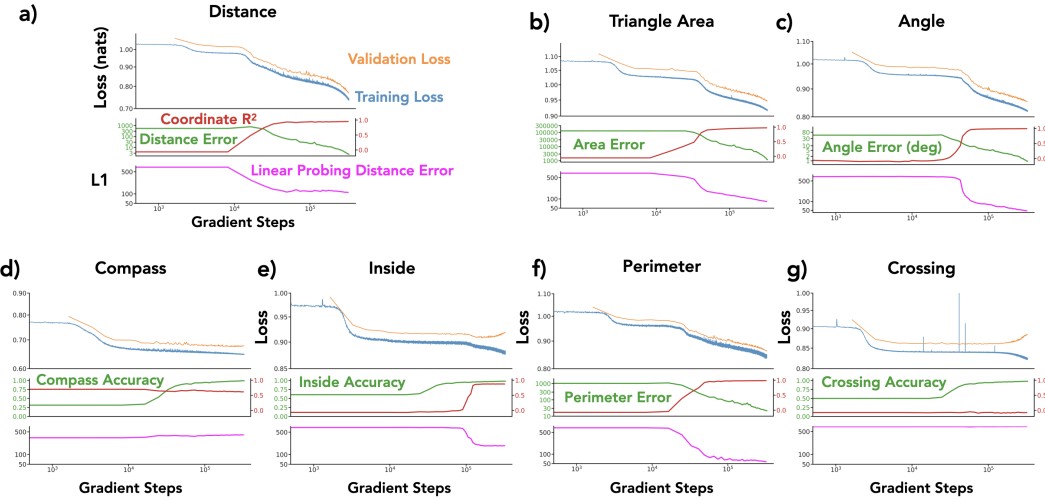

Figure 6: **Training dynamics for all single-task models.** (a) `distance`, (b) `trianglearea`, (c) `angle`, (d) `compass`, (e) `inside`, (f) `perimeter`, (g) `crossing`. Each panel shows three rows: (top) training loss (blue) and validation loss (orange); (middle) task-specific metric (green, left axis) and linear probe coordinate $R^2$ (red, right axis); (bottom) linear probing distance error (magenta). All plots use log-scale x-axis for gradient steps.

**Representation Dynamics.** Fig. 7 visualizes how internal representations evolve during training via PCA projections at six checkpoints. A striking pattern emerges: once a representational structure

forms, it remains largely fixed throughout the subsequent training phase where task accuracy continues to improve. Examining the gradient steps, representations are essentially fixed in the first ~15% of training, remaining static while loss slowly decreases and accuracy rises. The `distance` task (top row) establishes its thread-like structure early; `angle` (middle row) settles into a 2D manifold; `compass` (bottom row) forms fragmented regional clusters, all within the first few checkpoints, with minimal subsequent change. What determines when representations stop evolving remains unclear, though it appears correlated with the initial loss drop. This may relate to recently observed gradient dynamics in language model training, where loss deceleration phases exhibit qualitatively different learning behavior (Mircea et al., 2025).

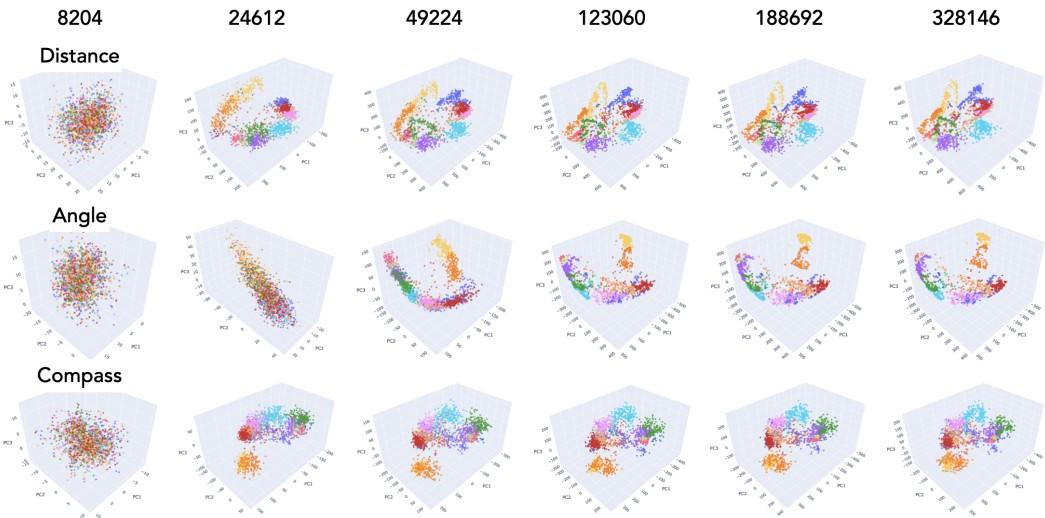

Figure 7: **Representation dynamics during training.** Rows: `distance` (top), `angle` (middle), `compass` (bottom). Columns show PCA projections at gradient steps 8204, 24612, 49224, 123060, 188692, and 328146 (left to right). Cities are colored by geographic region.

### E.2 QUALITATIVE REPRESENTATIONS

Fig. 8 shows PCA projections of city representations for single-task models across three random seeds (rows). The `distance` task consistently produces characteristic thread-like structures. `Angle` and `perimeter` often form larger 2D manifold-like structures. `triangle area` tends to produce arc-shaped geometries. `Compass` forms local clusters corresponding to directional categories, while `inside` produces a more global, diffuse structure.

While there is some seed-to-seed variability within each task, the broader categories remain distinguishable: `distance` representations are qualitatively distinct from the cluster-based representations of `compass` and `inside`, and both differ from the manifold-like structures produced by `triangle area`, `angle`, and `perimeter`.

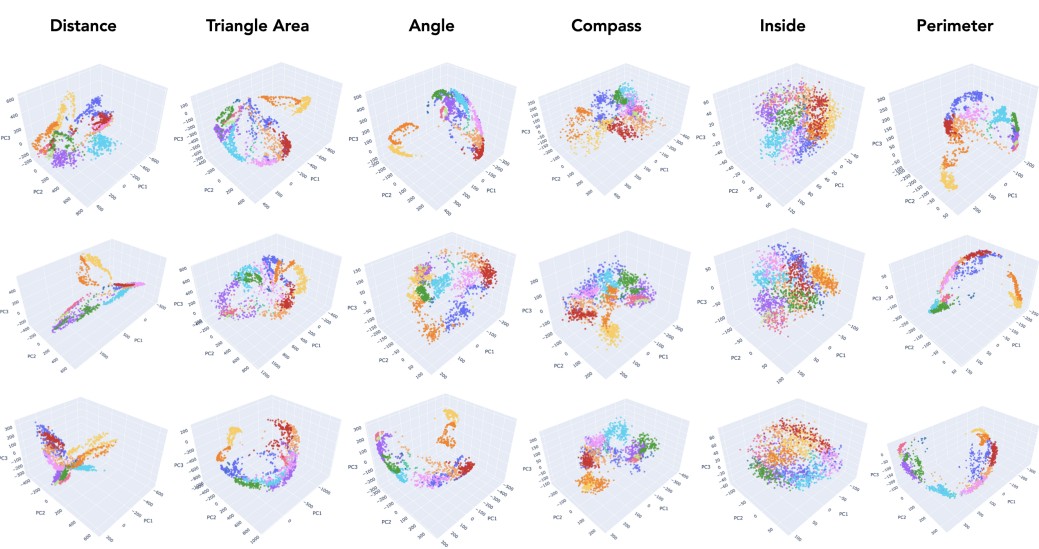

Figure 8: **Representation visualizations for single-task models across multiple seeds.** Each column shows a different task; each row shows a different random seed. Cities are colored by geographic region. Despite seed variability, task-specific geometric patterns are visible.

### E.3 ADDITIONAL CKA RESULTS

**Single-Task CKA Across Layers.** Fig. 9 shows CKA matrices for single-task models at layers 3, 4, 5, and 6. Each cell shows mean ± SEM across 3 seeds. We observe: (1) CKA values increase from layer 3 to layers 4–6, indicating that world representations become more consistent in later layers; (2) the `distance` task (D) shows lower CKA with other tasks across all layers, consistent with its divergent representational geometry; (3) `crossing` (Cr) shows near-zero CKA due to training failure in single-task settings; (4) diagonal entries (same task) can show significant variability, indicating that even identical training objectives can yield different representational solutions.

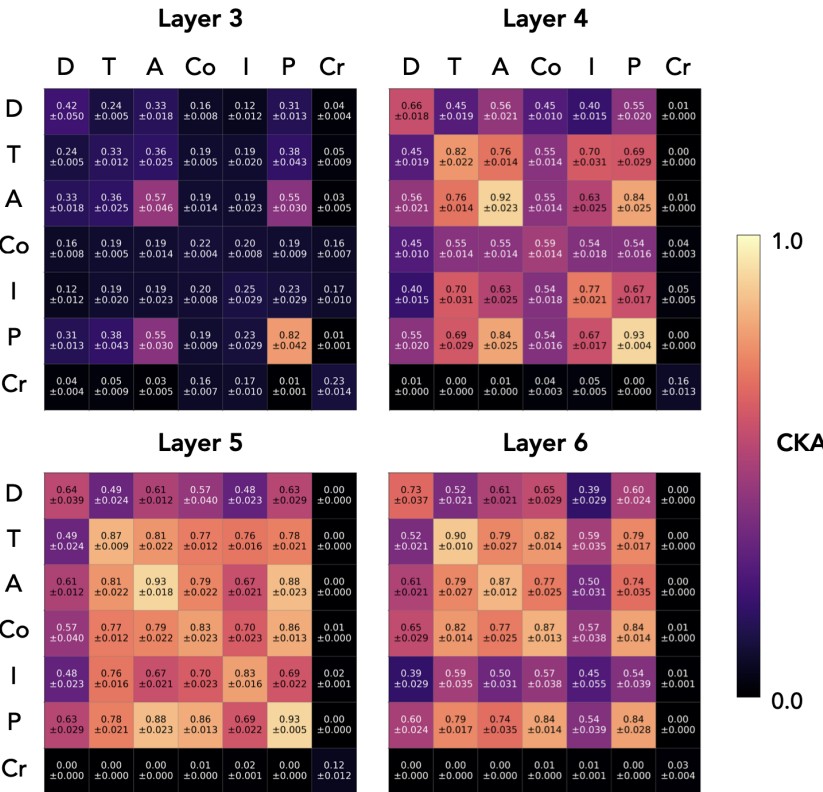

Figure 9: **CKA matrices for single-task models across layers.** Each cell shows mean ± SEM across 3 seeds. D=distance, T=triangle area, A=angle, Co=compass, I=inside, P=perimeter, Cr=crossing. CKA increases in later layers; `distance` shows consistently lower cross-task similarity.

**Two-Task CKA.** Fig. 10 shows the CKA matrix for two-task models at layer 5. Compared to single-task models (Fig. 9, layer 5), two-task training substantially increases representational alignment: all off-diagonal entries exceed 0.84, compared to values as low as 0.48 for single-task models. Notably, diagonal entries (same task combination, different seeds) show minimum CKA of 0.89, indicating that multi-task training also reduces inter-seed variance. For diagonal entries, we exclude same-seed comparisons (which trivially yield 1.0) and report only the upper triangle since the matrix is symmetric. This confirms the main text finding that multi-task training drives representational convergence.

**CKA vs. Task Count (Per-Seed).** Fig. 11 shows the same CKA vs. task count analysis as Fig. 3(d) in the main text, but broken down by individual seeds. Each panel shows one seed. These per-seed values are pooled to produce the main text figure, where error bars represent SEM across seeds. The pattern is consistent across all three seeds: CKA increases substantially from 1 to 2 tasks and saturates at 2–3 tasks for layers 4–6.

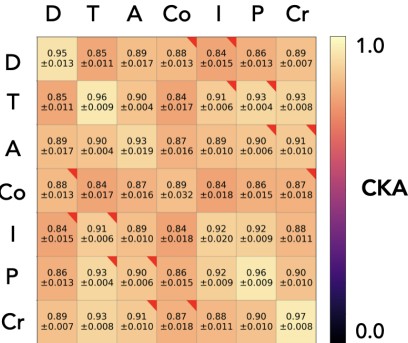

Figure 10: **CKA matrix for two-task models at layer 5.** Mean $\pm$ SEM across 3 seeds. All pairs show high alignment ($>0.84$), substantially higher than single-task models.

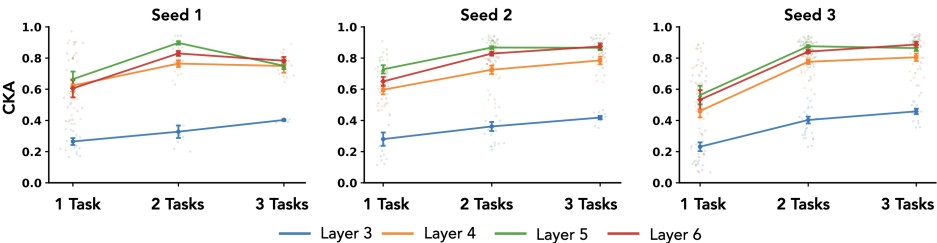

Figure 11: **CKA vs. task count for individual seeds.** Each panel shows a different seed. These values are pooled in Fig. 3(d); error bars there represent SEM across seeds.

**Aggregated CKA Trends.** Fig. 12(a) shows CKA vs. task count for a single seed, using all $\binom{7}{2} = 21$ two-task models and all $\binom{7}{3} = 35$ three-task models, but only comparing non-overlapping pairs (models sharing no common tasks). This yields 105 non-overlapping pairs for 2-task models and 70 for 3-task models. Fig. 12(b) shows within-task CKA (same task combination, different seeds) as a function of task count, demonstrating that multi-task training also reduces seed-to-seed variability: representations become more consistent not just across tasks but also across random initializations.

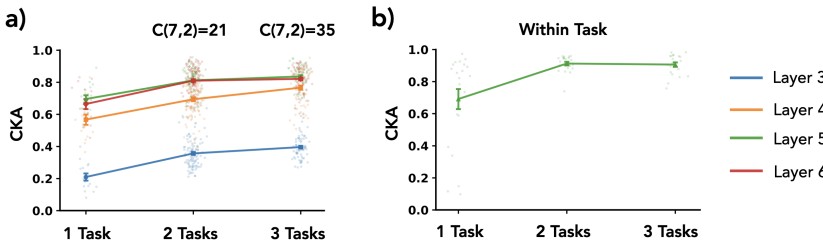

Figure 12: **Aggregated CKA analysis.** (a) CKA vs. task count for single seed, comparing only non-overlapping model pairs (105 pairs for 2-task, 70 pairs for 3-task). (b) Within-task CKA (same task combination, different seeds) increases with task count, indicating multi-task training reduces seed variability.

