# OpenReview forum: "Multi-Task Pretraining Drives Representational Convergence"
_ICLR.cc/2026/Workshop/Sci4DL — Sci4DL 2026_

### Official Review · Reviewer_PFWR · 2026-02-24

**Fit:** 3
**Significance:** 2
**Confidence:** 3

**Summary:**

This paper investigates how task composition shapes internal representations in neural networks, providing controlled evidence for the Multitask Scaling Hypothesis (MSH), a proposed mechanism underlying the Platonic Representation Hypothesis. The authors train small transformers on 7 geometric tasks defined over real-world city coordinates (5,075 cities), decoupling the "world" from the data generation process. Key findings include:

1. Representations crystallize early (~15% of training) and remain frozen thereafter
2. Different tasks produce distinct geometries
3. Multi-task training drives convergence: models trained on different task combinations show higher CKA alignment, even with completely disjoint tasks
4. A 7-task model spontaneously recovers world-map structure in raw PCA**—without requiring a linear probe
5. The "crossing" task fails to train alone but succeeds with multi-task scaffolding

**Strengths:**

### Excellent Workshop Fit
The paper directly addresses the workshop's focus on "scientific methods for understanding deep learning." It takes a principled, controlled approach to studying representation formation—exactly the kind of methodological contribution the workshop seeks.

### Clever Experimental Design
The framework decoupling world, data, and model is well-conceived. By using real city coordinates as ground truth and defining multiple geometric tasks over them, the authors isolate the effect of task diversity on representation geometry. This controlled setup avoids confounds present in studying PRH in large-scale models.

### Clear Evidence for the Multitask Scaling Hypothesis
The CKA results convincingly show that:
- Single-task models produce variable, task-specific representations
- Multi-task models converge toward similar representations
- This holds even for models trained on completely disjoint task sets

This provides concrete, testable evidence for a specific mechanism proposed in the PRH literature.

### PCA Finding
The observation that the 7-task model's raw PCA reveals world-map structure (without probing) is compelling. It suggests that multi-task training doesn't just create *decodable* representations—it amplifies them until they dominate the variance structure. This is a nice concrete demonstration of how task diversity shapes representation geometry.

### Interesting Auxiliary Findings
- The "crossing" task failing alone but succeeding with multi-task training suggests implicit curriculum effects through shared representations
- Early crystallization of representations (~15% of training) connects to recent work on loss plateaus and phase transitions in learning

**Suggestions:**

### 1. Clarify Training Objective in Main Text
The autoregressive next-token prediction objective and model architecture (Qwen2) are only described in Appendix C. Moving a brief description to the main text would help readers build intuition for why an internal representation of city coordinates emerges during training.

### 2. Consider Additional Similarity Metrics
CKA is invariant to orthogonal transformations, meaning high CKA indicates shared relational structure but not necessarily identical geometry. Adding a shape metric like **Procrustes distance** ([Williams et al., NeurIPS 2021](https://proceedings.neurips.cc/paper/2021/hash/252a3dbaeb32e7690242ad3b556e626b-Abstract.html)) would reveal whether multi-task training drives representations toward the same geometric solution or just similar relational structures.

### 3. Address Architecture Dependence
The paper tests only one architecture (Qwen2, 128 hidden dim, 6 layers). The PRH concerns convergence across architectures. While testing the task-diversity mechanism is valuable, briefly acknowledging this limitation and/or including one additional architecture (even a simple baseline) would strengthen the claims.

### 4. Discuss the Compass Task's Early Stability
Looking at Figures 6-7, the compass task appears to achieve high coordinate R² and stable representational geometry earlier than other tasks. This suggests categorical/directional tasks may impose different learning dynamics. Discussing why some tasks induce faster representational crystallization would add insight.

### 5. Minor Clarifications
- Figure 3(d): What do the dots represent? (Individual seeds? Task combinations?)
- The paper would benefit from briefly explaining what "autoregressive training" means for readers less familiar with transformer terminology
- Use the full name "Centered Kernel Alignment (CKA)" at first mention in the abstract.

---

### Official Review · Reviewer_zyxG · 2026-02-27

**Fit:** 3
**Significance:** 2
**Confidence:** 2

**Summary:**

This study documents the dependency of a networks representational geometry on its task and finds that representations become more similar when networks are trained on multiple non-overlapping tasks. Representational similarity is quantified by CKA on transformer networks trained autoregressively on several geometric tasks relating the location of cities on a two dimensional world map.

**Strengths:**

Studying the representational structure of neural networks can reveal the algorithms they use to solve tasks, bringing us one step closer to interpretable mechanistic explanation for their performance. The simple synthetic tasks studied here offer a controlled test bed for exploring the effect of task demands on the learned representation.

**Suggestions:**

Have you considered running control experiments matching network capacity with task demand (eg. varying width)? Would smaller networks trained on single task align more with one another than large under-constrained ones do?
How robust are your findings to the choice of a representational similarity measure, and how strong is the evidence for convergence (how does it vary with capacity/width)?
How would you study the weight similarity structure underlying the representational geometry?
How is this representation of a world map used by subsequent layers of the network?

---

### Official Review · Reviewer_9xZQ · 2026-02-27

**Fit:** 2
**Significance:** 2
**Confidence:** 2

**Summary:**

The representations produced by a transformer is studied in the context of multi-task training. It is observed that single task training produces diverse representations when the task is varied. Conversely, multi-task training produces convergent representations when these tasks are varied. It is shown, to a certain extent, that multi-task training amplifies the magnitude of linear world representations, until they dominate the principal components. The authors claim that these findings provide partial evidence for the Platonic Representation Hypothesis.

**Strengths:**

The paper is concise and makes a clear point on how the study provides circumstantial evidence for the Platonic Representation Hypothesis. The methodology is well described and the experimental steps are easy to follow as a reader. The authors did a good job in designing the experiment such that they can illustrate their point.

**Suggestions:**

The figures are often hard to read, I would suggest including less figures and decluttering them.
Additionally, some of the results could be merged. For example, result 1 does not really stand on its own and is used more to set the stage for the following results, which the authors focus on.
The strength of the claims seems disproportionate with the results that are presented. It is not immediately clear how the results really support the Platonic Representation Hypothesis. The authors could show this more directly.

---

### Meta-Review · Area_Chair_nif3 · 2026-02-28

**Recommendation:** Accept

**Metareview:**

This paper shows in a controlled setting that training on multiple tasks lead to a convergence of the geometry of representations. All reviewers noted that the presented experiments validate the hypothesis. I recommend acceptance.

---

### Decision · Program_Chairs · 2026-03-02

Accept